# The Genesis of Pyrite in the Fule Pb-Zn Deposit, Northeast Yunnan Province, China: Evidence from Mineral Chemistry and In Situ Sulfur Isotope

**Meng Chen, Tao Ren * and Shenjin Guan ***

Faculty of Land and Resources Engineering, Kunming University of Science and Technology, Kunming 650093, China
* Correspondence: rentao@kust.edu.cn (T.R.); guansj@kust.edu.cn (S.G.)

**Abstract:** The Fule deposit is a typical Cd-, Ge- and Ga-enriched Pb-Zn deposit located in the southeast of the Sichuan–Yunnan–Guizhou Pb-Zn polymetallic ore province in China. Zoned, euhedral cubic and pentagonal dodecahedral and anhedral pyrites were observed, and they are thought to comprise two generations. First generation pyrite (Py1) is homogeneous and entirely confined to a crystal core, whereas second generation pyrite (Py2) forms bright and irregular rims around the former. Second generation pyrite also occurs as a cubic and pentagonal dodecahedral crystal in/near the ore body or as an anhedral crystal generally closed to the surrounding rock. The content of S, Fe, Co, and Ni in Py1 are from 52.49 to 53.40%, 41.91 to 44.85%, 0.19 to 0.50% and 0.76 to 1.55%, respectively. The values of Co/Ni, Cu/Ni and Zn/Ni are from 0.22 to 0.42, 0.02 to 0.08 and 0.43 to 1.49, respectively, showing that the Py1 was formed in the sedimentary diagenetic stage. However, the contents of S, Fe, Co, and Ni in Py2 are in the range from 51.67 to 54.60%, 45.01 to 46.52%, 0.03 to 0.07% and 0.01 to 0.16%, respectively. The Co/Ni, Cu/Ni and Zn/Ni values of Py2 are from 0.40 to 12.33, 0.14 to 13.70 and 0.04 to 74.75, respectively, which is characterized by hydrothermal pyrite (mineralization stage). The different $\delta^{34}S$ values of the Py1 ($-34.9$ to $-32.3‰$) and the Py2 (9.7 to 20.5‰) indicate that there are at least two different sources of sulfur in the Fule deposit. The sulfur in Py1 was derived from the bacterial sulfate reduction (BSR), whereas the sulfur in the ore-forming fluids (Py2) was derived from the thermochemical sulfate reduction (TSR). The main reasons for the different morphologies of pyrite in the regular spatial distribution in the Fule deposit are temperature and sulfur fugacity.

**Keywords:** in situ S isotope; elemental geochemistry; pyrite; Fule Pb-Zn deposit; northeast Yunnan; China

## 1. Introduction

The Sichuan–Yunnan–Guizhou (SYG) polymetallic ore concentration area in the southwestern margin of the Yangtze block is an essential part of the low-temperature metallogenic domain in South China and one of the major production bases of Pb-Zn-Ag, and sphalerite contains significant amounts of trace elements, including Ge, Ga and Cd [1–3]. More than five hundred Pb-Zn polymetallic deposits have been produced in the region [1–3].

The Fule deposit has a mining history spanning more than 300 years. It is representative of the many large-scale lead–zinc deposits in the Sichuan–Yunnan–Guizhou Pb-Zn polymetallic metallogenic province. Since 1955, extensive research has been conducted on the deposit, including analysis of the ore field structure [1], the trace elements enrichment mechanism [4,5], the characteristics and evolution of the ore-forming fluids [5], the source of the ore-forming materials [6,7], and the metallogenic chronology [8,9]. Although much research work has been determined, there are still controversies in understanding the age of the ore formation, the mechanism of ore formation and the type of deposit. Some researchers believe that the deposit is MVT, while others believe that it is genetically related

to Emeishan basalt [6–8]. Most researchers have the opinion that the deposit was formed in the late Indosinian period (191.9~222 Ma) [8,9], while some researchers consider that it was formed in the Himalayan period (20.4~34.7 Ma) [8,9].

Pyrite is one of the most abundant minerals in various deposits. More and more studies show that pyrite, with a complex internal structure and morphology, often records mineralization information [10–13]. Therefore, the study of pyrite can be used not only to reconstruct the hydrothermal evolution process [14], but also to define the genesis of the deposit [10,13–16]. Previous studies on pyrite have mainly involved the Carlin-type gold deposit [15,17,18], the epithermal deposit [19], the porphyry copper deposit [16] and the VMS-type deposit [20]. However, only the Huize deposit in the SYG polymetallic area was investigated for pyrite genesis [21,22].

This work found zoned, euhedral, and anhedral pyrites in the Fule deposit. As mentioned, the pyrites' major element contents were analyzed by electron microprobe. In addition, the in situ sulfur isotope analyses of pyrite were carried out by LA-MC-ICP-MS, and the environment, sulfur sources and genesis of pyrites were discussed.

## 2. Geological Background

The Fule is a large-sized Pb-Zn deposit with high Cd, Ge, and Ga contents (Figure 1a; [6,7,22]). The exploration results show that the lead and zinc metal reserves of the Fule deposit are 0.6 Mt. The deposit contains very high ore grades (up to 60 wt% Zn + Pb, average 15–20 wt%) [23]. In addition, the deposit contains metal reserves of approximately 4567 t Cd, 329 t Ge and 177 t Ga, with average grades of 0.127 wt% Cd, 0.012 wt% Ge and 0.007 wt% Ga, respectively [6].

The Permian Yangxin Formation ($P_2y$) is an ore-host stratum in the deposit, mainly composed of dolomite intercalated with limestone. It can be divided into the following three lithological sections. The lower section ($P_2y^1$) is composed of light gray limestone (Figure 1b). The middle section ($P_2y^2$) consists mainly of light gray limestone interbedded with dolomite, and locally contains siliceous dolomite, which is the main host rock of the deposit. The upper section ($P_2y^3$) is composed of gray medium-thick layered crystalline limestone, and a small amount of dolomitic limestone with chert strips. The igneous rocks in the area are Emeishan basalt, which is a series of continental rift tholeiite assemblages containing dense massive basalt and basaltic tuff. The main structures in the mining area are the Tuoniu-Duza anticline and the Mile-Shizong fault [22]. Together, they control the distribution of regional strata, secondary structures and mineralization. The Tuoniu-Duza anticline has a flat shape with a dip angle of 10° to 12° [22].

It mainly consists of three ore blocks: Laojuntai, Xinjuntai and Tonniu. The Fule deposit is buried about 150 m~200 m below the surface. Currently, 28 lead and zinc orebodies have been delineated with the NE strike, with the dip angle of 10° in the SE (Figure 1c) extending more than 3000 m [22]. The orebodies occur as stratiform to lentiform shapes or as veins along fractures within the Yangxin Formation. Metallic minerals in the deposit mainly include sphalerite, galena, pyrite, and a little chalcopyrite, tetrahedrite, tennantite, millerite, vaesite, gersdorffite and polydymite. Secondary oxides include cerussite and malachite.

According to the mineral assemblages and in combination with the previously published geological data [22], the ore-forming process of the Fule deposit can be divided into diagenetic and hydrothermal periods. The hydrothermal period can be further divided into sulfide-and-carbonate and carbonate stages.

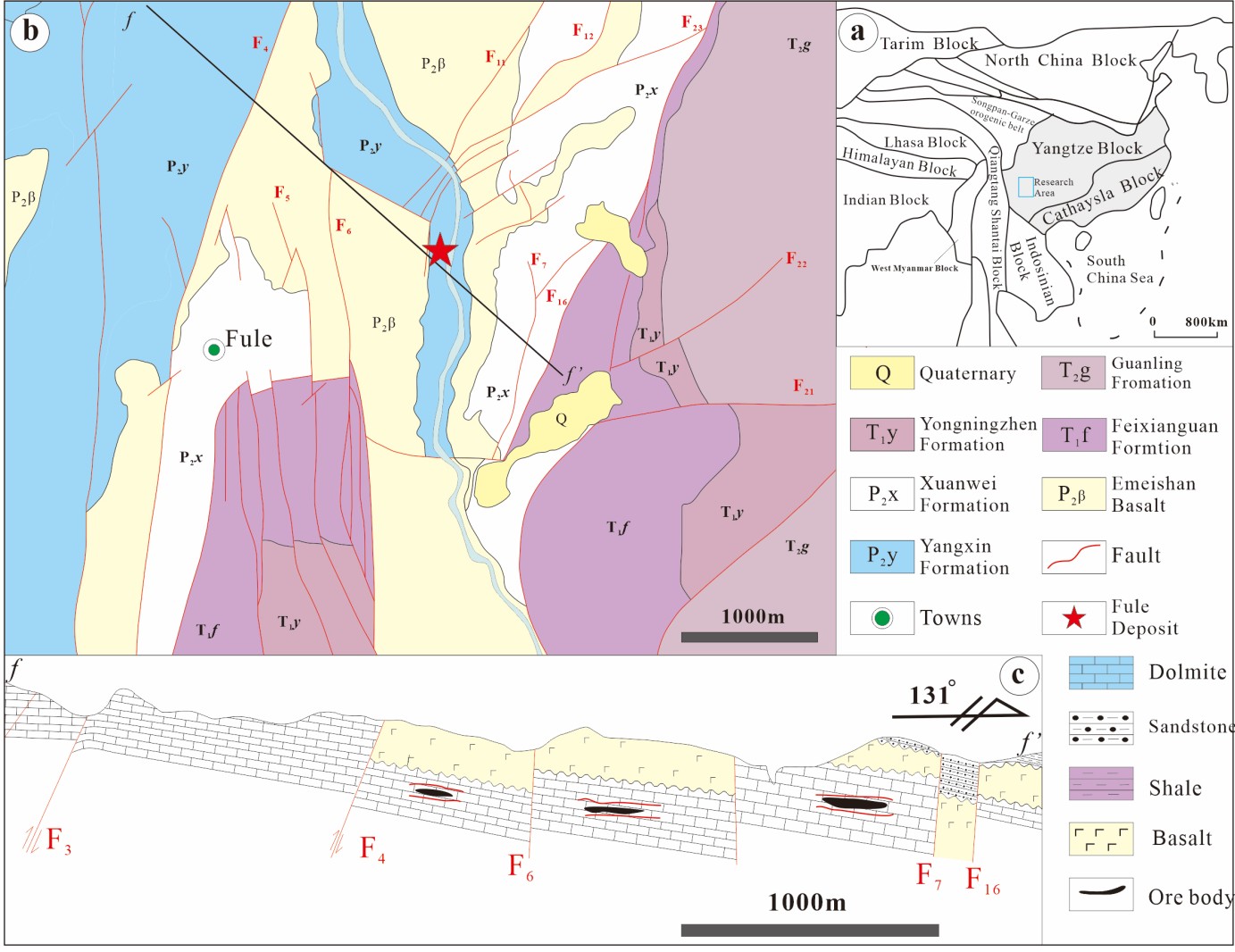

**Figure 1.** Regional geological setting of SW China (**a**), geological sketch map (**b**) and f-f' geological section (**c**) of the Fule Pb-Zn deposit (after Ref. [22]).

## 3. Sampling and Analytical Methods

All samples in this study were collected from the 1440 level of the Fule Pb-Zn deposit. Representative samples were selected for major elements and in situ sulfur isotope composition analysis. The in situ major element analysis of pyrite was carried out in the State Key Laboratory of Geochemistry, Institute of Geochemistry, Chinese Academy of Sciences. The instrument was a JXA-8230 electron probe, with an acceleration voltage of 25 kV, a current of 10 nA and a beam spot diameter of 1–5 μm. The SPI#02753-AB was used as the standard sample. The detection limits were from 100 to 200 ppm. Precisions for major elements and trace elements were approximately ±2% and ±10%, respectively. In situ sulfur isotope analysis was performed using a Nu plasma II multi receiver inductively coupled plasma mass spectrometer (MC-ICP-MS) equipped with a resolution S-155 Ar-F laser ablation system with a resolution of 193 nm in the GPMR laboratory of the China University of Geosciences (Wuhan). Working conditions included a laser energy density of 3 J/cm$^2$, a spot diameter of 33 μm and a single-point ablation time of 40 s. Natural pyrite WS-1 ($\delta^{34}S_{V-CDT} = 1.1 \pm 0.2‰$) was used to calibrate the sulfur isotope deviation, and V-CDT (Vienna-Cañon Diablo troilite) was used as a standard for the measured sulfur isotope data ($\delta^{34}S$).

## 4. Mineralogical Characteristics of Pyrite

Based on the morphology and chemical composition, pyrites are zoned and of two generations, with the first generation occupying the crystal core (Py1), whereas the other forms rims (Py2). Py2 formed in the second stage is intimately related to Pb-Zn mineralization, including cubic, pentagonal dodecahedral and anhedral crystals (Figure 2).

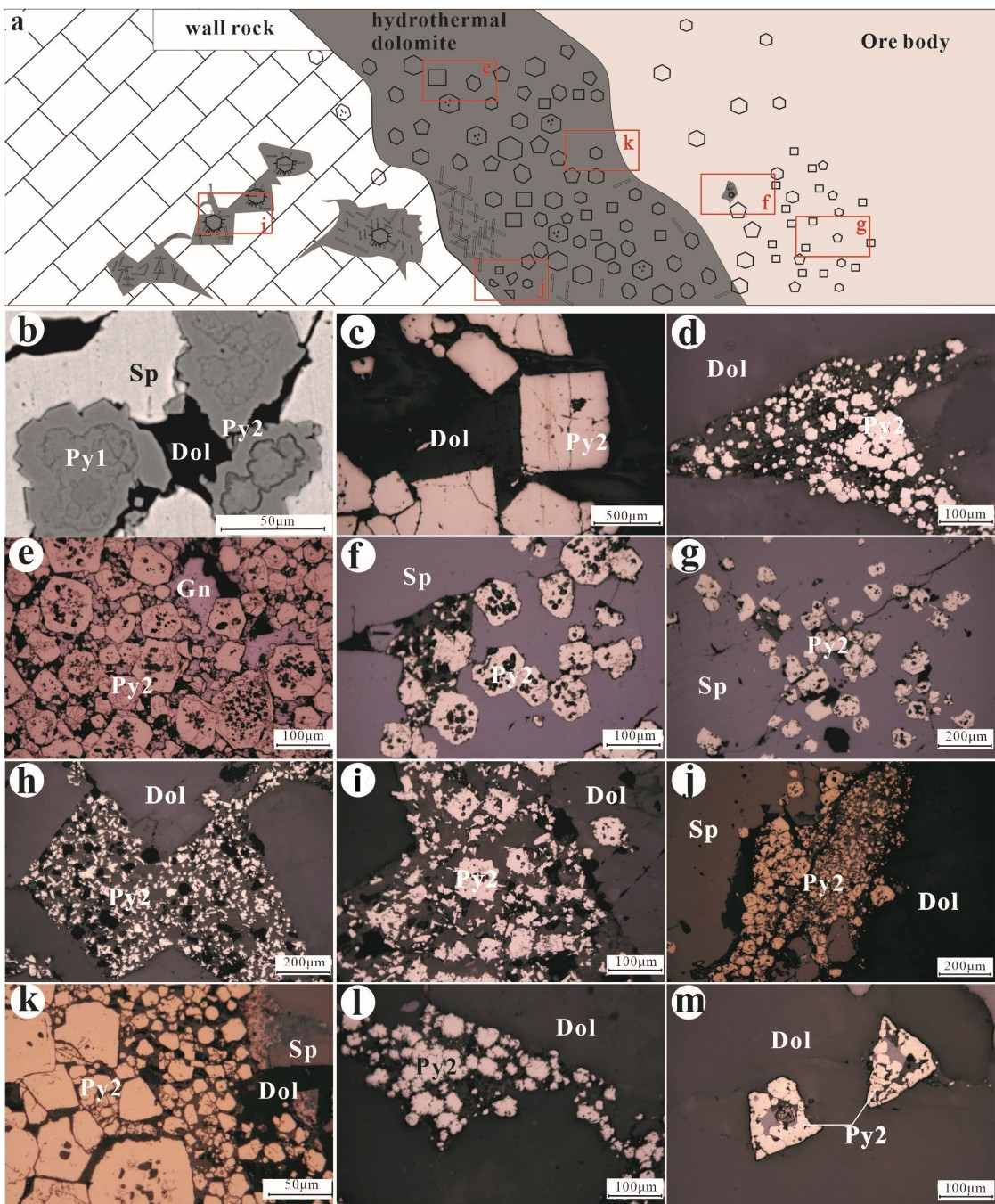

**Figure 2.** Mineral assemblage sketch (**a**) and microscopic images of pyrite (**b**–**m**) in the Fule deposit. (**b**) Zoned pyrite contains a Py1 core and a Py2 mantle; (**c**) cubic pyrite in the hydrothermal dolomite vein; (**d**) pentagonal dodecahedral pyrite filled in a fissure; (**e**,**f**) euhedral and subhedral pyrite containing subrounded dolomite, black voids in the pyrite are dolomite; (**g**) subhedral pyrite wrapped in sphalerite; (**h**,**i**) anhedral pyrite filled in the cavity; (**j**,**k**) anhedral pyrite aggregates occurring at the contact zone of sphalerite and dolomite; (**l**) granular pyrite aggregates; (**m**) intergrowth of pyrite and sphalerite filled in the cavity of dolomite; Abbreviations: Gn—galena; Sp—sphalerite; Py—pyrite; Dol—dolomite.

(1) Zoned pyrite (Figure 2b) consists of a homogeneous core (Py1) and a bright mantle (Py2) with a particle size of 50–80 μm.

(2) Euhedral pyrite includes pentagonal dodecahedral and cubic crystals (Figure 2c,d) with significant variations in crystal size, distributed in the center of the hydrothermal dolomite vein (Figure 2a). Pentagonal dodecahedral pyrite is also present in the fracture of recrystallized dolomite or the contact with sphalerite (Figure 2d). The large cubic crystal occurs in the fissure of recrystallized dolomite (Figure 2c). A small amount of cubic crystal occurs at the margin of the hydrothermal dolomite vein or is enveloped in sphalerite.

(3) Subhedral euhedral pyrite occurs at the margin of the pyrite-bearing hydrothermal dolomite vein or filled in the fissure of recrystallized dolomite (Figure 2e–g).

(4) Anhedral pyrite (Figure 2h–l) occurs in the hydrothermal dolomite vein near the recrystallized dolomite (Figure 2l,m).

## 5. Results

### 5.1. Major Elements

The results of the electron microprobe analysis of pyrite are shown in Table 1. The variation ranges of S and Fe in Py1 are from 52.49 to 53.40% and from 41.91 to 44.85%, respectively. Py1 is rich in Co, Ni and As, with contents of 0.19~0.50%, 0.76~1.55% and 0.44~1.37%, respectively. The S content (51.67 to 54.60%) of Py2 is slightly lower than that of Py1. Py2 has an Fe content of 45.01 to 46.52%. Py2 is relatively rich in Pb and Se, with values of 0.02~1.61% and 0.01~0.09%, respectively. The contents of Co, Ni and As in Py2 are 0.01 to 0.16%, 0.03 to 0.07% and 0.01 to 0.65%, respectively.

### 5.2. In Situ S Isotope Analysis

The results of the in situ S isotope analysis of pyrite are presented in Table 2 and Figure 3. Sulfur isotope values of pyrite in the Fule deposit are from -34.9 to 20.5‰ ($n = 29$). The $\delta^{34}S$ values in Py1 are from $-34.9$ to $-32.3$‰ ($n = 2$), while in Py2 they are from 9.7 to 20.5‰ ($n = 26$). The $\delta^{34}S$ values gradually increased from 10.9 to 13.6‰ from the core to the rim in a cubic pyrite crystal (Figure 4).

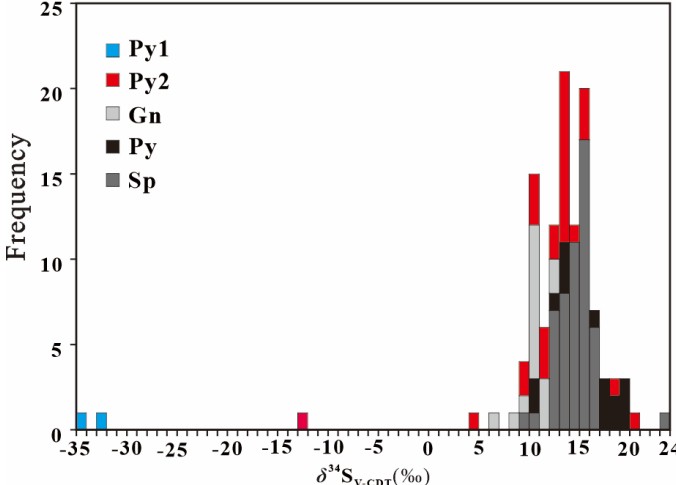

**Figure 3.** Histogram of the sulfur isotopic compositions of sulfide minerals from the Fule deposit. Data are taken from Refs. [7,9,22] and this paper. Abbreviations: Gn—galena; Sp—sphalerite; Py—pyrite; Dol—dolomite.

**Table 1.** Chemical composition of pyrite in the Fule deposit.

| Sample No. | Pyrite Type | | Fe | Cu | Zn | Ni | Se | As | S | Cd | Pb | Co | Total | Co/Ni | Cu/Ni | Zn/Ni | S/Fe |
|---|---|---|---|---|---|---|---|---|---|---|---|---|---|---|---|---|---|
| | | | % | | | | | | | | | | | | | | |
| fl17-3-core | | | 44.85 | 0.06 | 1.13 | 0.76 | 0.00 | 0.44 | 53.03 | 0.01 | 0.01 | 0.19 | 100.48 | 0.25 | 0.08 | 1.49 | 2.06 |
| fl17-4-core | | | 44.29 | 0.07 | 1.09 | 1.29 | 0.03 | 0.99 | 52.49 | 0.01 | 0.04 | 0.28 | 100.59 | 0.22 | 0.06 | 0.85 | 2.06 |
| fl17-5-core | | | 41.91 | 0.04 | 0.66 | 1.55 | 0.01 | 1.37 | 52.96 | - | - | 0.50 | 99.00 | 0.32 | 0.03 | 0.43 | 1.10 |
| fl17-6-core | Py1 | Zoned pyrite | 42.15 | 0.02 | 0.48 | 1.05 | 0.06 | 1.14 | 53.40 | - | - | 0.45 | 98.76 | 0.42 | 0.02 | 0.46 | 1.10 |
| S.D. | | | 0.28 | 0.01 | 0.02 | 0.27 | 0.02 | 0.28 | 0.27 | - | 0.02 | 0.05 | 0.05 | 0.02 | 0.01 | 0.32 | 0.00 |
| Median | | | 44.57 | 0.07 | 1.11 | 1.03 | 0.02 | 0.72 | 52.76 | 0.01 | 0.03 | 0.24 | 100.54 | 0.24 | 0.07 | 1.17 | 2.06 |
| Mean | | | 44.57 | 0.07 | 1.11 | 1.03 | 0.02 | 0.72 | 52.76 | 0.01 | 0.03 | 0.24 | 100.54 | 0.24 | 0.07 | 1.17 | 2.06 |
| fl2001-02 | | | 46.20 | 0.05 | 0.10 | 0.08 | 0.04 | 0.38 | 53.32 | 0.02 | 0.03 | 0.06 | 100.30 | 0.78 | 0.58 | 1.28 | 2.01 |
| fl2001-03 | | | 46.15 | 0.01 | 0.09 | - | 0.04 | 0.19 | 53.28 | 0.03 | 0.02 | 0.06 | 99.85 | - | - | - | 2.01 |
| fl2001-09 | | | 46.41 | - | 0.02 | 0.01 | 0.03 | 0.43 | 53.11 | 0.01 | - | 0.04 | 100.05 | 3.89 | - | 1.67 | 1.99 |
| fl2001-10 | | | 46.30 | - | 0.79 | - | 0.05 | 0.52 | 53.67 | - | - | 0.05 | 101.38 | - | - | - | 2.02 |
| fl2001-11 | | | 45.95 | - | 2.01 | - | - | 0.18 | 54.03 | 0.01 | - | 0.05 | 102.23 | - | - | - | 2.05 |
| fl2001-12 | | | 45.01 | - | 1.86 | - | 0.03 | 0.32 | 52.27 | 0.01 | - | 0.05 | 99.55 | - | - | - | 2.02 |
| fl2013-17 | | | 46.45 | - | 0.14 | 0.02 | - | 0.23 | 53.80 | - | - | 0.03 | 100.66 | 1.67 | - | 7.72 | 2.02 |
| fl2013-14 | | | 46.29 | 0.03 | 0.36 | - | 0.09 | 0.25 | 53.73 | - | 0.02 | 0.04 | 100.81 | - | - | - | 2.02 |
| fl2001-04 | | | 45.64 | 0.07 | 0.05 | 0.13 | 0.05 | 0.65 | 52.83 | 0.01 | 0.20 | 0.05 | 99.68 | 0.40 | 0.52 | 0.35 | 2.02 |
| fl2001-05 | | Subhedral-euhedral pyrite | 45.88 | 0.08 | 0.35 | 0.03 | - | 0.21 | 53.66 | 0.02 | - | 0.06 | 100.26 | 1.96 | 2.71 | 12.46 | 2.04 |
| fl2001-07 | Py2 | | 46.27 | 0.03 | 0.15 | - | - | 0.09 | 53.69 | - | 0.13 | 0.07 | 100.41 | - | - | - | 2.02 |
| fl2001-08 | | | 45.64 | 0.04 | 0.05 | - | - | - | 52.12 | 0.01 | 0.15 | 0.04 | 98.05 | 12.33 | 13.67 | 18.00 | 1.99 |
| fl2013-15 | | | 45.84 | 0.01 | 1.06 | - | 0.01 | - | 53.62 | - | 0.09 | 0.03 | 100.65 | - | - | - | 2.04 |
| fl2013-19 | | | 46.11 | 0.04 | 0.27 | - | - | 0.59 | 53.46 | 0.01 | 0.05 | 0.06 | 100.57 | - | - | - | 2.02 |
| fl2013-20 | | | 46.27 | 0.06 | 0.05 | 0.04 | - | 0.11 | 53.50 | 0.01 | - | 0.06 | 100.08 | 1.63 | 1.69 | 1.34 | 2.01 |
| fl2013-22 | | | 46.20 | 0.01 | - | - | 0.01 | 0.13 | 53.59 | 0.02 | - | 0.06 | 100.02 | - | - | - | 2.02 |
| fl2013-24 | | | 46.13 | - | 0.01 | - | - | 0.27 | 53.19 | - | - | 0.05 | 99.65 | - | - | - | 2.01 |
| fl2013-25 | | | 46.08 | 0.03 | - | 0.04 | - | 0.29 | 53.47 | - | - | 0.05 | 99.96 | 1.27 | 0.76 | 0.11 | 2.02 |
| fl2013-30 | | | 46.03 | 0.04 | - | - | 0.01 | - | 54.60 | - | 0.06 | 0.05 | 100.78 | - | - | - | 2.07 |
| S.D. | | | 0.33 | 0.02 | 0.63 | 0.04 | 0.02 | 0.16 | 0.55 | 0.01 | 0.06 | 0.01 | 0.82 | 3.66 | 4.69 | 6.27 | 0.02 |
| Median | | | 46.13 | 0.04 | 0.15 | 0.04 | 0.04 | 0.26 | 53.50 | 0.01 | 0.06 | 0.05 | 100.26 | 1.65 | 1.23 | 1.51 | 2.02 |
| Mean | | | 46.04 | 0.04 | 0.46 | 0.05 | 0.04 | 0.30 | 53.42 | 0.01 | 0.08 | 0.05 | 100.26 | 2.99 | 3.32 | 5.37 | 2.02 |

**Table 1.** *Cont.*

| Sample No. | Pyrite Type | Fe | Cu | Zn | Ni | Se | As | S | Cd | Pb | Co | Total | Co/Ni | Cu/Ni | Zn/Ni | S/Fe |
|---|---|---|---|---|---|---|---|---|---|---|---|---|---|---|---|---|
| | | | | | | | % | | | | | | | | | |
| fl10-01 | | 46.52 | 0.03 | 0.05 | - | 0.03 | - | 53.59 | - | - | 0.05 | 100.27 | - | - | - | 2.01 |
| fl10-02 | | 45.92 | 0.14 | 0.08 | 0.01 | - | 0.04 | 53.76 | - | - | 0.06 | 100.01 | 6.30 | 13.70 | 8.00 | 2.04 |
| fl10-03 | | 46.18 | 0.09 | 0.72 | - | 0.03 | 0.01 | 53.81 | - | - | 0.06 | 100.89 | - | - | - | 2.03 |
| fl2002-01 | | 45.91 | 0.02 | - | - | - | 0.13 | 52.93 | 0.01 | 0.07 | 0.05 | 99.12 | - | - | - | 2.01 |
| fl2002-02 | | 46.40 | - | 0.01 | - | - | - | 53.25 | - | 0.05 | 0.06 | 99.75 | - | - | - | 2.00 |
| fl2002-03 | Anhedral pyrite | 45.66 | 0.04 | 0.02 | 0.02 | - | 0.10 | 51.67 | - | - | 0.05 | 97.56 | 2.70 | 1.75 | 1.10 | 1.97 |
| fl2002-04 | | 45.84 | 0.01 | 0.02 | - | - | 0.03 | 53.33 | - | - | 0.04 | 99.28 | - | - | - | 2.03 |
| fl17-10 | | 45.68 | 0.33 | 0.02 | - | - | - | 53.08 | - | 0.07 | 0.05 | 99.23 | - | - | - | 2.02 |
| fl17-11 | | 45.06 | 0.29 | 0.18 | - | - | - | 51.85 | 0.01 | 1.61 | 0.06 | 99.06 | - | - | - | 2.00 |
| fl2001-01 | | 46.24 | 0.03 | 0.02 | - | - | - | 54.07 | - | 0.07 | - | 100.44 | - | - | - | 2.04 |
| S.D. | | 0.40 | 0.11 | 0.22 | 0.01 | - | 0.05 | 0.76 | - | 0.62 | 0.01 | 0.89 | 1.80 | 5.98 | 3.45 | 0.02 |
| Median | Py2 | 45.92 | 0.04 | 0.02 | 0.02 | 0.03 | 0.04 | 53.29 | 0.01 | 0.07 | 0.05 | 99.52 | 4.50 | 7.73 | 4.55 | 2.02 |
| Mean | | 45.94 | 0.11 | 0.12 | 0.02 | 0.03 | 0.06 | 53.13 | 0.01 | 0.37 | 0.05 | 99.56 | 4.50 | 7.73 | 4.55 | 2.02 |
| fl2001-13 | | 45.96 | 0.02 | 0.01 | 0.16 | - | 0.43 | 53.25 | 0.01 | 0.03 | 0.07 | 99.92 | 0.44 | 0.14 | 0.04 | 2.02 |
| fl2013-18 | Pentagonal dodecahedral pyrite | 45.58 | 0.00 | 0.39 | - | 0.04 | 0.52 | 52.22 | 0.03 | 0.02 | 0.05 | 98.88 | - | - | - | 2.00 |
| S.D. | | 0.19 | 0.01 | 0.19 | 0.00 | 0.00 | 0.05 | 0.50 | 0.01 | 0.01 | 0.01 | 0.52 | 0.00 | 0.00 | 0.00 | 0.01 |
| Median | | 45.77 | 0.01 | 0.20 | 0.16 | 0.04 | 0.48 | 52.76 | 0.02 | 0.03 | 0.06 | 99.40 | 0.44 | 0.14 | 0.04 | 2.01 |
| Mean | | 45.77 | 0.01 | 0.20 | 0.16 | 0.04 | 0.48 | 52.76 | 0.02 | 0.03 | 0.06 | 99.40 | 0.44 | 0.14 | 0.04 | 2.01 |
| fl17-1-rim | | 45.78 | 0.00 | 1.50 | 0.02 | - | 0.08 | 53.41 | 0.03 | - | 0.05 | 100.87 | 2.30 | 0.20 | 74.75 | 2.03 |
| fl17-2-rim | Zoned pyrite | 45.54 | 0.03 | 1.63 | 0.05 | - | 0.10 | 53.96 | - | 0.04 | 0.05 | 101.41 | 0.98 | 0.47 | 30.81 | 2.06 |
| S.D. | | 0.12 | 0.02 | 0.06 | 0.02 | - | 0.01 | 0.28 | 0.00 | 0.00 | 0.00 | 0.27 | 0.66 | 0.14 | 21.97 | 0.02 |
| Median | | 45.66 | 0.02 | 1.57 | 0.04 | - | 0.09 | 53.69 | 0.03 | 0.04 | 0.05 | 101.14 | 1.64 | 0.34 | 52.78 | 2.05 |
| Mean | | 45.66 | 0.02 | 1.57 | 0.04 | - | 0.09 | 53.69 | 0.03 | 0.04 | 0.05 | 101.14 | 1.64 | 0.34 | 52.78 | 2.05 |

**Table 2.** In situ LA-MC-ICP-MS sulfur isotopic composition of pyrite from the Fule deposit (‰).

| Sample No. | Description | Pyrite Type | $\delta^{34}$S |
|---|---|---|---|
| fl17-2-core | | Py1 | −32.9 |
| fl-17-3-core | Zoned pyrite | | −34.3 |
| fl17-1-rim | | | −12.5 |
| fl-20-1-rim | | | 13.6 |
| fl-20-2-rim | | | 12.0 |
| fl-20-3-core | Cubic pyrite | | 10.9 |
| fl-20-4-rim | | | 13.0 |
| fl-20-5-rim | | | 13.6 |
| fl-20-6-rim | | | 13.1 |
| fl20-13-8 | | | 13.6 |
| fl20-13-1 | Pentagonal dodecahedral pyrite | | 13.5 |
| fl20-13-3 | | | 14.0 |
| fl20-13-12 | | Py2 | 9.8 |
| fl20-13-13 | | | 9.7 |
| fl20-13-6 | | | 10.0 |
| fl20-13-5 | | | 11.1 |
| fl20-13-2 | | | 12.1 |
| fl20-13-4 | Subhedral-euhedral pyrite | | 10.5 |
| fl20-13-7 | | | 11.3 |
| fl20-13-9 | | | 13.7 |
| fl20-13-10 | | | 13.3 |
| fl20-13-11 | | | 15.0 |
| fl17-8 | | | 11.4 |
| fl10-5 | | | 13.4 |
| Fl17-2 | | | 18.1 |
| fl17-3 | Anhedral pyrite | | 13.1 |
| fl17-4 | | | 15.4 |
| fl17-5 | | | 15.2 |
| fl17-6 | | | 20.5 |

## 6. Discussion

### 6.1. Chemical Composition and Sulfur Sources of Pyrite

Pyrite contains more than 30 kinds of trace elements, including chalcophile, lithophile and siderophile elements [10–13]. The trace element contents are closely related to the type and genetic type of the deposit, as well as the temperature and pressure conditions [24]. Pyrites formed in high-temperature hydrothermal deposits are generally rich in siderophile and lithophile elements. They also have high amounts of Bi, Cu, Zn and As. Under moderate temperatures, pyrite is mainly rich in Cu, Au, Pb, Zn, Bi, Ag, etc. Pyrite in epithermal deposit has high Hg, Sb, Ag and As content. Compared to other types of deposits, the pyrite in the Fule deposit is characterized by the enrichment of Cu, As, Co, Ni and Se, and the contents of most trace elements are low, indicating that the deposit was formed under medium-low temperature conditions. It is consistent with the homogenization temperature of fluid inclusion in Fule sphalerite [5]. Cu, Ni and Co enrichment in Py2 and Cd, Ge and Ga enrichment in sphalerite indicate that the deposit may be an MVT deposit. Ore bodies host in dolomite and have simple mineral assemblages (mainly of sphalerite, galena and pyrite), which are also basically consistent with the geological characteristics of typical MVT deposits.

Many sulfur isotope analyses have been conducted on sphalerite, galena and pyrite in the Fule deposit, and the $\delta^{34}$S values are concentrated from 10.04 to 19.30‰ [7,22]. This indicates that there is a seawater sulfate reservoir existing in the strata, and thermochemical sulfate reduction (TSR) is mainly the formation mechanism of reduced sulfur in the deposit [10]. However, some researchers still believe that there may be multiple sulfur sources [7,22]. In this study, Py1 with $\delta^{34}$S of −34.9‰ to −32.3‰ was first found in the Fule

deposit. Therefore, based on the previous research and the sulfur isotope results obtained in this work, sulfur in the Fule deposit has at least two different sources (Figure 4).

TSR and bacterial sulfate reduction (BSR) are the two main mechanisms for the conversion of sulfate to reduced sulfur [25]. $H_2S$ of TSR origin inherits the sulfur isotopic composition of sulfate, and the resulting sulfide generally has high sulfur isotopic value [26] and a relatively high formation temperature (100 to 140 °C). The Py2 formed in the metallogenic stage has a $\delta^{34}S$ of 10 to 20‰, which is similar to that of sphalerite and galena reported in previous studies [22]. This isotopic value is close to the Permian, Carboniferous and Cambrian marine sulfate (11‰, 14‰ and 17‰, respectively, [22]) in northeastern Yunnan, indicating that the sulfate in the sedimentary strata is the main source of sulfur. TSR may be the main mechanism for reducing sulfur formation in the ore-forming stage.

BSR can occur near the surface or in shallow burial environments at low temperatures (<80 °C; [27–29]). BSR can result in significant isotopic fractionation, generally between 4 to 46‰, and 65 ‰ in extreme cases [30,31], such that $H_2S$ produced by BSR has sulfur isotope values of −50‰ and 30‰ [31]. The $\delta^{34}S$ of the Py1 in the Fule deposit ranges from −34.9 to −32.3‰. This value is about 40‰ lower than that of the hydrothermal sulfide, indicating that the sulfur in Py1 is of BSR origin.

The $\delta^{34}S$ values of the cubic pyrite (Py2) show a gradual increase from the core to the rim. According to previous studies, when sulfide precipitated in a hydrothermal solution dominated by $H_2S$ and containing a small amount of $SO_4^{2-}$, its $\delta^{34}S$ value was similar to that of the initial solution in the early stage, but higher than that of the initial solution in the late phase [32]. This may be the reason for the change in the sulfur isotope of cubic pyrite in the Fule deposit.

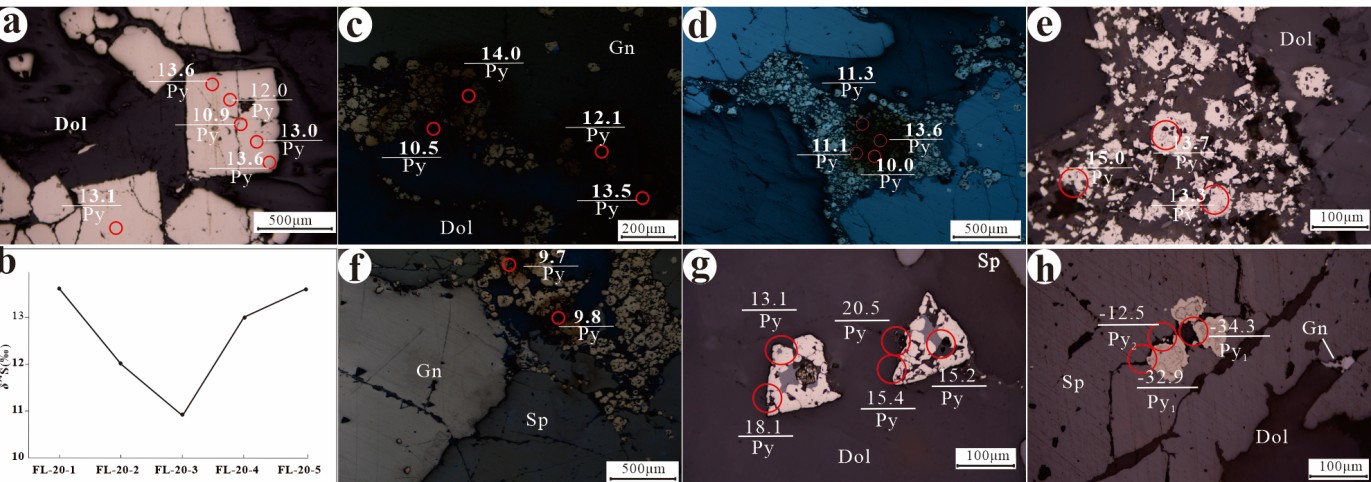

**Figure 4.** Photographs showing in situ LA-MC-ICP-MS sulfur isotope analysis spots and $\delta^{34}S$ values. (**a**) cubic pyrite in hydrothermal dolomite; (**b**) the sulfur isotope values in the cubic pyrite core are relatively lower than that of rim; (**c**,**d**) anhedral pyrite aggregate; (**e**) anhedral pyrite filled in the cavity; (**f**) subhedral–anhedral pyrite filled in the contact zone of galena and sphalerite; (**g**) intimate growth of pyrite and sphalerite; (**h**) zoned pyrite has a core and a mantle filled in the fissure of sphalerite. Abbreviations: Gn—galena; Sp—sphalerite; Py—pyrite; Dol—dolomite.

## 6.2. Genesis of Pyrite

The Co/Ni ratio in pyrite could be used to effectively indicate the genesis of pyrite [33,34]. The contents of Co and Ni in sedimentary pyrite are generally low and the Co/Ni values are less than 1. The Co and Ni contents and Co/Ni values of hydrothermal pyrite vary greatly, with 1 < Co/Ni < 5. Pyrite formed in volcanic exhalative massive sulfide deposits has high Co and low Ni contents and high Co/Ni values (5~50) [34–36].

Py1 in the Fule deposit had relatively higher Co and Ni contents than Py2. In addition, the content of Ni was much higher than that of Co, and the Co/Ni values were between 0.22

and 0.42, which indicate that Py1 was formed during the depositional period (deposited at the same time as the ore-hosted rock). However, the Co/Ni ratio in Py2 was between 0.40 and 12.33 (mostly are from 1 to 5), which is consistent with the value of hydrothermal pyrite. The Cu/Ni and Zn/Ni values from Py1 in the Fule deposit were from 0.02 to 0.08 and from 0.43 to 1.49, respectively. These values plot in the range of sedimentary pyrite (0.01 to 2 and 0.01 to 10) [37,38]. In contrast, the ratios from Py2 ranged from 0.14 to 13.70 and 0.04 to 74.75, also indicating a hydrothermal origin.

A systematic study of pyrite formed under natural and synthetic conditions showed that temperature and (or) sulfur saturation really influenced the pyrite morphology. The crystal appearance experienced a change trend from columnar → cube → pentagonal dodecahedron or octahedron → irregular granular [39]. As mentioned above, hydrothermal pyrite in the Fule deposit had a significant morphological zone in space (Figure 2a). Columnar pyrite distributed in the margin of hydrothermal dolomite near the wall rock shows that the crystal selects the {100} and extends growth rapidly along the <001> direction. It can be formed in a large temperature gradient and low material conditions. Meanwhile, cubic and pentagonal dodecahedral pyrites appear inside the ore body or in the center of hydrothermal dolomite due to suitable temperature, sufficient material supply and high sulfur fugacity.

## 7. Conclusions

(1) Pyrite in the Fule deposit has various morphologies, including zoned, euhedral (cubic and pentagonal dodecahedral), subhedral-euhedral and anhedral crystals. The pyrite core was formed during the sedimentary stage, and the rim, euhedral and anhedral pyrite was formed in the metallogenic stage.

(2) The high S/Fe ratio, As, Cu, and Zn contents reflect that Py2 was formed in a medium-low temperature environment. The sulfur isotopic composition of the pyrite in the deposit shows that there are at least two different sulfur sources. The sulfur in Py1 was derived from BSR, while the sulfur in Py2 was derived from the TSR.

(3) Pyrite morphology in the Fule deposit goes through the changing trends of columnar, cube, pentagonal dodecahedron, and irregular granular morphology. Temperature and (or) sulfur saturation dominate the morphological change.

**Author Contributions:** Data curation, investigation, writing—original draft, M.C.; conceptualization, methodology, writing—review and editing, T.R. and S.G. All authors have read and agreed to the published version of the manuscript.

**Funding:** This study was supported by the National Natural Science Foundation of China (NSFC) project (42163005).

**Data Availability Statement:** Data are contained within the article.

**Acknowledgments:** We would like to thank Zhaojun Lv for the assistance during fieldwork.

**Conflicts of Interest:** The authors declare no conflict of interest.

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
