# Peer review of "The Genesis of Pyrite in the Fule Pb-Zn Deposit, Northeast Yunnan Province, China: Evidence from Mineral Chemistry and In Situ Sulfur Isotope"

_minerals, doi:10.3390/min13040495_

Round 1

Reviewer 1 Report

The ore deposit is Mississippi-Valley type (MVT)? If yes, please explain in the text.

The legend from figure 1b should be the same as the one from 1c. It is not known to which formations in 1b the rocks in figure 1c correspond. Please make the correlation between figures 1b and 1c. If the rocks in figure 1c (dolomite, sandstone, shale) belong to the P2y Yangxin Formation, then it must be specified in figure 1c or the symbols for these rocks (dolomite, sandstone, shale) must be colored blue.

Regarding the genesis of pyrite, please discuss it in correlation with the genesis of the deposit. If the deposit falls under the Mississippi-Valley type (MVT), then the discussion about the genesis must be in agreement with this type (MVT). This type of deposit would be suggested by δ34S for pyrite 1 and formation temperatures.

Author Response

1# This paper focused on pyrite, so a short review of pyrite should be summarized in the Introduction part.

We have summarized a short review about pyrite in “Introduction” part.

2# Line37:How large it is ? lead and zinc reserves ?

We have added the content in the text. Lead-zinc reserves of the Fule deposit are 0.6 Mt. In addition, the deposit contains proven reserves of approximately 4567t Cd, 329t Ge, and 177t Ga.

3# Line51:much more information are needed for this deposit, such as the reserves, ore types etc. what are they ? 

We have added this part, see “2 Geological Background” part.

4# please clarify the subordination relationship between the characteristics and types of pyrite

We have clarified the relationship between pyrite morphology and mineral paragenetic sequence, see “4. Mineralogical Characteristics of Pyrite” part.

5# Line140:“standard pyrite values (S=53.45%, Fe=46.55%)” references are needed 

We have deleted the expression.

6# Line144(5.1(3)): It seems doesn't show these trend in Fig.3.

Due to instrument detection limit, the changing trend of elements is not clearly reflected in the mapping.

7# The ore deposit is Mississippi-Valley type (MVT)? If yes, please explain in the text.

The legend from figure 1b should be the same as the one from 1c. It is not known to which formations in 1b the rocks in figure 1c correspond. Please make the correlation between figures 1b and 1c. If the rocks in figure 1c (dolomite, sandstone, shale) belong to the Py2 Yangxin Formation, then it must be specified in figure 1c or the symbols for these rocks (dolomite, sandstone, shale) must be colored blue.

Regarding the genesis of pyrite, please discuss it in correlation with the genesis of the deposit. If the deposit falls under the Mississippi-Valley type (MVT), then the discussion about the genesis must be in agreement with this type (MVT). This type of deposit would be suggested by δ34S for pyrite 1 and formation temperatures.

We believe that the deposit is a MVT deposit, which has been briefly supplemented in the text.

We have modified Figure 1 accordingly.

Reviewer 2 Report

1.       English needs to be improved, including but not limited to grammar throughout the manuscript and consistency.

2.       This paper focused on pyrite, so a short review of pyrite should be summarized in the Introduction part.

3.       The Figure of paper is too zoomed and too small such as Fig. 2. Very difficult to see something in these tiny photos where observing microscopic and small detail is often key to evaluate the texture of a mineral.

4.       Much more introduction of the deposit is needed such as metal reserves and ore-forming stages.

5.       Other comments please see the pdf file of the MS.

Author Response

Reviewer #2

1#: English needs to be improved, including but not limited to grammar throughout the manuscript and consistency.

Thank you for your suggestion. We have carried out a comprehensive language polish.

2#: This paper focused on pyrite, so a short review of pyrite should be summarized in the Introduction part.

We have reviewed about pyrite in “Introduction” part.

3#: The Figure of paper is too zoomed and too small such as Fig. 2. Very difficult to see something in these tiny photos where observing microscopic and small detail is often key to evaluate the texture of a mineral.

We have corrected accordingly.

4#: Much more introduction of the deposit is needed such as metal reserves and ore-forming stages.

We have added related content.

5#: Other comments please see the pdf file of the MS.

Thank you very much for your detailed comments. We have corrected accordingly.

Reviewer 3 Report

The paper by Chen et al. is about morphology and chemistry of various pyrite types from Fule Pb-Zn Deposit, Northeast Yunnan Province, China. The paper is of interest, but should be modified before the acceptance.

The main comment is related to an inaccurate presentation of the material: look at the pdf file, which contains many remarks, especially, in section Results.

Do not use obsolete terms (idiomorphic, xenomorphic), especially, because you mix them with most common euhedral/subhedral terms. Do not integrate Discussion moments in section Results (e.g., origin of pyrite).

Remove elemental maps with rock-forming elements from Fig. 3 and replace them by maps with Co, Ni and As or other necessary trace elements of pyrite

The presentation of chemical analyses requires statistical parameters: average, standard deviation, median, otherwise, you comparison is invalid. Some contents of trace elements are below detection limit. You must indicate it Analytical section and remove insignificant values from Table.

Two analyses of Py1 are insufficient for Discussion even if they are similar.

Since the chemical composition of pyrite in section Results is followed by S isotopy, the same logic should be presented in section Discussion.

Abstract and Conclusions should be modified according to modifications in paper.

Author Response

Reviewer #3

1# Line 33, 34, 39, 51: “dispersed elements” should be “trace elements”.

Strictly speaking, dispersed elements are a kind of trace elements, but this paper we focus on Ga, Ge, and Cd. These elements are also called dispersed elements.

2# Line 67: “Them” should be “They”

We have modified.

3# what is the detection limit for elements? You show 0.1wt % Ni or Cd in pyrite: I doubt that these results are meaningful

The detection limits in this study are 100-200 ppm.

4# Line 81: “current of 10 nA” should be “a current of 10 nA”, “...beam spot...”should be “a beam spot...”,

We have corrected accordingly.

5# Line 84: “equipped with Oxford... ” should be “ equipped with an Oxford... ”,

We have corrected accordingly.

6# Line 85: “...by Nu plasma ... ” should be “ ...by a Nu plasma ... ”,

We have corrected accordingly.

7# Line 87: “...with resolution S-155 ... ” should be “ ...with a resolution S-155 ... ”,

We have corrected accordingly.

8# Line 93-101:

Line 93: Figure e cannot occur prior to Figure a in Fig.2. wrong labels on figures? Your mineral assemblage is labeled by letter "e", check please accurately.

â‘ “idiomorphic and semi-idiomorphic...” in the description of picture (b,c) should be “ ... euhedral and subhedra ... ” and “...occurs in the contact zone of sphalerite...”in the description of picture (i) should be “ ... at the contact zone of sphalerite ... ”

â‘¡ “pyritohedron pyrite” of (f) is poor phrase

â‘¢ No lenticular dolomite in figures (b, c) can be seen.

â‘£ ask a question: what do you mean under "columnar"? I do not any columns in the description of picture (g, h)

⑤ in the description of picture (i): 1) I would rather use anhedral rather than xenomorphic (obsolete term) .  2) But these are not anhedral aggregates! These are interstitial aggregates of Py2 crystals

 â‘¥in the description of picture (l): I do not understand "irregular" term. I see granular pyrite aggregates, locally, with crystal shapes 

We have rearranged Figure 2.

We have corrected accordingly.

9# Line 102-128:  â‘ We describe that the Zoned pyrite of core py1 is “a homogeneous” in the Section 4(1) : how it can be homogeneous if it consists of many microcrystals?

Description of the formation of py1 and py2 during sedimentary-diagenesis and hydrothermal genesis, respectively: Out of place: this is Discussion. By the way, you should support your conclusions on the origin of pyrite.

â‘¡4(2) “...with large variations in crystal size. ” should come after “Euhedral pyrites include pentagonal dodecahedral and cubic pyrite ”(Line 112)ï¼›

“ Pentagonal dodecahedral pyrite also presents in the fracture of recrystallized dolomite or at the edge of sphalerite (Figure 2f).” should be “ Pentagonal dodecahedral pyrite is also present in the fracture of recrystallized dolomite or contact with sphalerite (Figure 2f)”(Line 113) . “The large grained cubic pyrite...” should be “cubic crystal”, cubic crystals - yes, cubic grains - NO (Line 114).

â‘¢4(3)Anhedral pyrite (irregular crystalline) is mainly filled in the intergranular porosity of recrystallized dolomite (Figure 2l, m). Few are present in the hydrothermal dolomite (Figure 2c). “irregular crystalline”should be deleted; “filled ”should be “occurs”;  “porosity ” should be“space”.

We have corrected accordingly.

10#: Fig 3:a-why do you Ca map for pyrite?

b-why do you need maps with rock-forming elements for pyrite? You should provide maps with Ni, Co, As, etc.!

Due to detection limit of scanning electron microscope, trace elements such as Ni may not be observed.

11#:  Line 132-147, Relevant suggestions on 5.1 Major elements.

Very detailed suggestions, thanks. We have corrected accordingly.

12#: â‘  5.2 In-situ S isotope, the title should be changed to“5.2 In-situ S isotope analysis ”; "is" should be changed to "are" in the first and second sentences. In the first sentence, "result" should be changed to"results". The third sentence should be deleted “Where”, and LA-MC-ICPMS is changed to LA-ICPMS in Table 2 and Fig. 4.

â‘¡The third sentence about Py1 questions and suggestions: Two values are insignificant for any discussions. Do you have only two grains of Py1? If you have possibility, I strongly advice to make more analyses of Py1 to obtain necessary statistic for conclusion

â‘¢grains have no crystal forms

cubic crystals - yes, cubic grains - NO

We have corrected accordingly. At present, we observe relatively small number of Py1.

13#: â‘  It's better first to discuss the chemical composition of pyrite rather than isotopes following the logic of Results

â‘¡ 6.1. Sulfur sources: For the article“TSR ” and “BSR ” , provide first the full term and then abbreviation

â‘¢ what do you mean“ metallogenic stage” in the sentence “The Py2 formed in the metallogenic stage has a δ34S of 10 to 20‰, which is similar to that of sphalerite and galena reported in previous studies [10]. ” ? unclear

â‘£ In Section 6.1, The last sentence of the second paragraph And the first sentence of the third paragraph are modified, "And" and "The" are deleted, and "with" in“ ...environments with low temperature” is changed to "at".

⑤ “The δ34S values of the cubic pyrite show a gradual increase from the core to the rim” in the first sentence of the last paragraph of Chapter 6.1. ,“ cubic pyrite ” must indicate that this is Py2

We have corrected accordingly. The sulfur isotope value in py2 is similar to that of sphalerite and galena in the deposit.

14#: 6.2 Genesis of pyrite

We have corrected accordingly.

15# Line133: “The result of electron microprobe analysis of pyrite is shown in Table 1”, “result ”should be “results”,and “is” should be delete.

We have corrected accordingly.

Round 2

Reviewer 3 Report

The authors made their best to accept the comments. There are a few poitns to be corrected in the manuscipt again (see attached pdf file).

Author Response

Response

Dear Reviewer,

Thank you very much for your comments. We have carefully read the comments and the manuscript has been extensively revised according to the comments one by one. All corrections are marked “in red” in the revised manuscript so that the reviewers and editors could easily identify the places of change.

Thank you and best regards.

The following is a point-to-point response to the three reviewers’ comments.

Reviewer #3

  1. Due to the scanning electron microscope has high detection limit, the Fig. 3 does not show meaningful geological information. So, we have deleted the Fig. 3.
  2. We have made a comprehensive modification to the table 1.
  3. We have revised the inaccurate words in the text.
  4. We have rearranged the references, and reduced the citation of Chinese articles. Only a few Chinese articles are cited in the parts of introduction and geological background.
